# Large Scale Adversarial Representation Learning

**Jeff Donahue**
DeepMind
jeffdonahue@google.com

**Karen Simonyan**
DeepMind
simonyan@google.com

## Abstract

Adversarially trained generative models (GANs) have recently achieved compelling image synthesis results. But despite early successes in using GANs for unsupervised representation learning, they have since been superseded by approaches based on self-supervision. In this work we show that progress in image generation quality translates to substantially improved representation learning performance. Our approach, BigBiGAN, builds upon the state-of-the-art BigGAN model, extending it to representation learning by adding an encoder and modifying the discriminator. We extensively evaluate the representation learning and generation capabilities of these BigBiGAN models, demonstrating that these generation-based models achieve the state of the art in unsupervised representation learning on ImageNet, as well as in unconditional image generation. Pretrained BigBiGAN models – including image generators and encoders – are available on TensorFlow Hub[1].

## 1 Introduction

In recent years we have seen rapid progress in generative models of visual data. While these models were previously confined to domains with single or few modes, simple structure, and low resolution, with advances in both modeling and hardware they have since gained the ability to convincingly generate complex, multimodal, high resolution image distributions [1, 17, 18].

Intuitively, the ability to generate data in a particular domain necessitates a high-level understanding of the semantics of said domain. This idea has long-standing appeal as raw data is both cheap – readily available in virtually infinite supply from sources like the Internet – and rich, with images comprising far more information than the class labels that typical discriminative machine learning models are trained to predict from them. Yet, while the progress in generative models has been undeniable, nagging questions persist: what semantics have these models learned, and how can they be leveraged for representation learning?

The dream of generation as a means of true understanding from raw data alone has hardly been realized. Instead, the most successful approaches for unsupervised learning leverage techniques adopted from the field of supervised learning, a class of methods known as *self-supervised* learning [4, 35, 32, 9]. These approaches typically involve changing or holding back certain aspects of the data in some way, and training a model to predict or generate aspects of the missing information. For example, [34, 35] proposed colorization as a means of unsupervised learning, where a model is given a subset of the color channels in an input image, and trained to predict the missing channels.

Generative models as a means of unsupervised learning offer an appealing alternative to self-supervised tasks in that they are trained to model the full data distribution without requiring any modification of the original data. One class of generative models that has been applied to representation learning is generative adversarial networks (GANs) [11]. The generator in the GAN

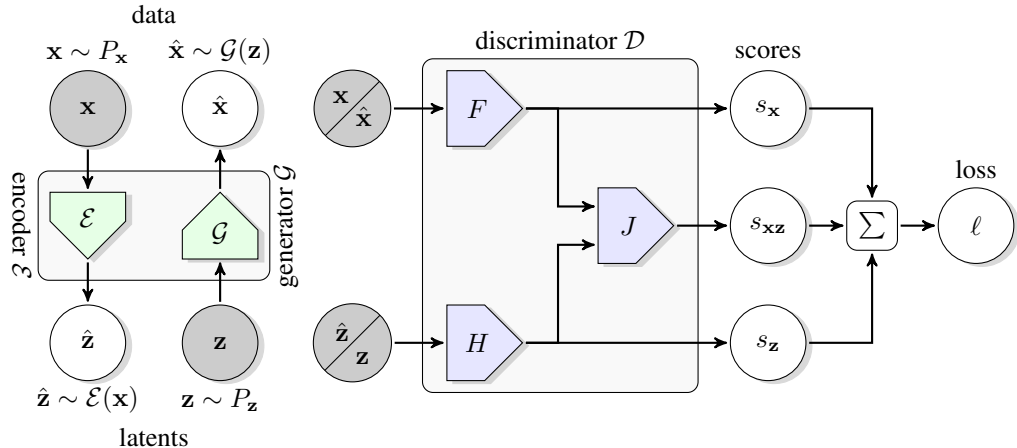

Figure 1: The structure of the BigBiGAN framework. The joint discriminator $\mathcal{D}$ is used to compute the loss $\ell$. Its inputs are data-latent pairs, either $(\mathbf{x} \sim P_{\mathbf{x}}, \hat{\mathbf{z}} \sim \mathcal{E}(\mathbf{x}))$, sampled from the data distribution $P_{\mathbf{x}}$ and encoder $\mathcal{E}$ outputs, or $(\hat{\mathbf{x}} \sim \mathcal{G}(\mathbf{z}), \mathbf{z} \sim P_{\mathbf{z}})$, sampled from the generator $\mathcal{G}$ outputs and the latent distribution $P_{\mathbf{z}}$. The loss $\ell$ includes the unary data term $s_{\mathbf{x}}$ and the unary latent term $s_{\mathbf{z}}$, as well as the joint term $s_{\mathbf{xz}}$ which ties the data and latent distributions.

framework is a feed-forward mapping from randomly sampled latent variables (also called "noise") to generated data, with learning signal provided by a *discriminator* trained to distinguish between real and generated data samples, guiding the generator's outputs to follow the data distribution. The *adversarially learned inference* (ALI) [8] or *bidirectional GAN* (BiGAN) [5] approaches were proposed as extensions to the GAN framework that augment the standard GAN with an *encoder* module mapping real data to latents, the inverse of the mapping learned by the generator.

In the limit of an optimal discriminator, [5] showed that a deterministic BiGAN behaves like an autoencoder minimizing $\ell_0$ reconstruction costs; however, the shape of the reconstruction error surface is dictated by a parametric discriminator, as opposed to simple pixel-level measures like the $\ell_2$ error. Since the discriminator is usually a powerful neural network, the hope is that it will induce an error surface which emphasizes "semantic" errors in reconstructions, rather than low-level details.

In [5] it was demonstrated that the encoder learned via the BiGAN or ALI framework is an effective means of visual representation learning on ImageNet for downstream tasks. However, it used a DCGAN [26] style generator, incapable of producing high-quality images on this dataset, so the semantics the encoder could model were in turn quite limited. In this work we revisit this approach using BigGAN [1] as the generator, a modern model that appears capable of capturing many of the modes and much of the structure present in ImageNet images. Our contributions are as follows:

- We show that BigBiGAN (BiGAN with BigGAN generator) matches the state of the art in unsupervised representation learning on ImageNet.

- We propose a more stable version of the joint discriminator for BigBiGAN.

- We perform a thorough empirical analysis and ablation study of model design choices.

- We show that the representation learning objective also improves unconditional image generation, and demonstrate state-of-the-art results in unconditional ImageNet generation.

- We open source pretrained BigBiGAN models on TensorFlow Hub[2].

## 2 BigBiGAN

The BiGAN [5] or ALI [8] approaches were proposed as extensions of the GAN [11] framework which enable the learning of an encoder that can be employed as an inference model [8] or feature representation [5]. Given a distribution $P_{\mathbf{x}}$ of data $\mathbf{x}$ (e.g., images), and a distribution $P_{\mathbf{z}}$ of latents $\mathbf{z}$

(usually a simple continuous distribution like an isotropic Gaussian $\mathcal{N}(0, I)$), the generator $\mathcal{G}$ models a conditional distribution $P(\mathbf{x}|\mathbf{z})$ of data $\mathbf{x}$ given latent inputs $\mathbf{z}$ sampled from the latent prior $P_{\mathbf{z}}$, as in the standard GAN generator [11]. The encoder $\mathcal{E}$ models the inverse conditional distribution $P(\mathbf{z}|\mathbf{x})$, predicting latents $\mathbf{z}$ given data $\mathbf{x}$ sampled from the data distribution $P_{\mathbf{x}}$.

Besides the addition of $\mathcal{E}$, the other modification to the GAN in the BiGAN framework is a joint discriminator $\mathcal{D}$, which takes as input data-latent pairs $(\mathbf{x}, \mathbf{z})$ (rather than just data $\mathbf{x}$ as in a standard GAN), and learns to discriminate between pairs from the data distribution and encoder, versus the generator and latent distribution. Concretely, its inputs are pairs $(\mathbf{x} \sim P_{\mathbf{x}}, \hat{\mathbf{z}} \sim \mathcal{E}(\mathbf{x}))$ and $(\hat{\mathbf{x}} \sim \mathcal{G}(\mathbf{z}), \mathbf{z} \sim P_{\mathbf{z}})$, and the goal of the $\mathcal{G}$ and $\mathcal{E}$ is to "fool" the discriminator by making the two joint distributions $P_{\mathbf{x}\mathcal{E}}$ and $P_{\mathcal{G}\mathbf{z}}$ from which these pairs are sampled indistinguishable. The adversarial minimax objective in [5, 8], analogous to that of the GAN framework [11], was defined as follows:

$$\min_{\mathcal{G}\mathcal{E}} \max_{\mathcal{D}} \left\{ \mathbb{E}_{\mathbf{x}\sim P_{\mathbf{x}}, \mathbf{z}\sim\mathcal{E}_{\Phi}(\mathbf{x})} \left[ \log(\sigma(\mathcal{D}(\mathbf{x}, \mathbf{z}))) \right] + \mathbb{E}_{\mathbf{z}\sim P_{\mathbf{z}}, \mathbf{x}\sim\mathcal{G}_{\Phi}(\mathbf{z})} \left[ \log(1 - \sigma(\mathcal{D}(\mathbf{x}, \mathbf{z}))) \right] \right\}$$

Under this objective, [5, 8] showed that with an optimal $\mathcal{D}$, $\mathcal{G}$ and $\mathcal{E}$ minimize the Jensen-Shannon divergence between the joint distributions $P_{\mathbf{x}\mathcal{E}}$ and $P_{\mathcal{G}\mathbf{z}}$, and therefore at the global optimum, the two joint distributions $P_{\mathbf{x}\mathcal{E}} = P_{\mathcal{G}\mathbf{z}}$ match, analogous to the results from standard GANs [11]. Furthermore, [5] showed that in the case where $\mathcal{E}$ and $\mathcal{G}$ are deterministic functions (i.e., the learned conditional distributions $P_{\mathcal{G}}(\mathbf{x}|\mathbf{z})$ and $P_{\mathcal{E}}(\mathbf{z}|\mathbf{x})$ are Dirac $\delta$ functions), these two functions are inverses at the global optimum: e.g., $\forall_{\mathbf{x}\in\text{supp}(P_{\mathbf{x}})} \mathbf{x} = \mathcal{G}(\mathcal{E}(\mathbf{x}))$, with the optimal joint discriminator effectively imposing $\ell_0$ reconstruction costs on $\mathbf{x}$ and $\mathbf{z}$.

While the crux of our approach, BigBiGAN, remains the same as that of BiGAN [5, 8], we have adopted the generator and discriminator architectures from the state-of-the-art BigGAN [1] generative image model. Beyond that, we have found that an improved discriminator structure leads to better representation learning results without compromising generation (Figure 1). Namely, in addition to the joint discriminator loss proposed in [5, 8] which ties the data and latent distributions together, we propose additional unary terms in the learning objective, which are functions only of either the data $\mathbf{x}$ or the latents $\mathbf{z}$. Although [5, 8] prove that the original BiGAN objective already enforces that the learnt joint distributions match at the global optimum, implying that the marginal distributions of $\mathbf{x}$ and $\mathbf{z}$ match as well, these unary terms intuitively guide optimization in the "right direction" by explicitly enforcing this property. For example, in the context of image generation, the unary loss term on $\mathbf{x}$ matches the original GAN objective and provides a learning signal which steers only the generator to match the image distribution independently of its latent inputs. (In our evaluation we will demonstrate empirically that the addition of these terms results in both improved generation and representation learning.)

Concretely, the discriminator loss $\mathcal{L}_{\mathcal{D}}$ and the encoder-generator loss $\mathcal{L}_{\mathcal{E}\mathcal{G}}$ are defined as follows, based on scalar discriminator "score" functions $s_*$ and the corresponding per-sample losses $\ell_*$:

$$
\begin{aligned}
s_{\mathbf{x}}(\mathbf{x}) &= \theta_{\mathbf{x}}^{\mathsf{T}} F_{\Theta}(\mathbf{x}) \\
s_{\mathbf{z}}(\mathbf{z}) &= \theta_{\mathbf{z}}^{\mathsf{T}} H_{\Theta}(\mathbf{z}) \\
s_{\mathbf{xz}}(\mathbf{x}, \mathbf{z}) &= \theta_{\mathbf{xz}}^{\mathsf{T}} J_{\Theta}(F_{\Theta}(\mathbf{x}), H_{\Theta}(\mathbf{z})) \\
\ell_{\mathcal{E}\mathcal{G}}(\mathbf{x}, \mathbf{z}, y) &= y\left(s_{\mathbf{x}}(\mathbf{x}) + s_{\mathbf{z}}(\mathbf{z}) + s_{\mathbf{xz}}(\mathbf{x}, \mathbf{z})\right) && y \in \{-1, +1\} \\
\mathcal{L}_{\mathcal{E}\mathcal{G}}(P_{\mathbf{x}}, P_{\mathbf{z}}) &= \mathbb{E}_{\mathbf{x}\sim P_{\mathbf{x}}, \hat{\mathbf{z}}\sim\mathcal{E}_{\Phi}(\mathbf{x})}\left[\ell_{\mathcal{E}\mathcal{G}}(\mathbf{x}, \hat{\mathbf{z}}, +1)\right] + \mathbb{E}_{\mathbf{z}\sim P_{\mathbf{z}}, \hat{\mathbf{x}}\sim\mathcal{G}_{\Phi}(\mathbf{z})}\left[\ell_{\mathcal{E}\mathcal{G}}(\hat{\mathbf{x}}, \mathbf{z}, -1)\right] \\
\ell_{\mathcal{D}}(\mathbf{x}, \mathbf{z}, y) &= h(ys_{\mathbf{x}}(\mathbf{x})) + h(ys_{\mathbf{z}}(\mathbf{z})) + h(ys_{\mathbf{xz}}(\mathbf{x}, \mathbf{z})) && y \in \{-1, +1\} \\
\mathcal{L}_{\mathcal{D}}(P_{\mathbf{x}}, P_{\mathbf{z}}) &= \mathbb{E}_{\mathbf{x}\sim P_{\mathbf{x}}, \hat{\mathbf{z}}\sim\mathcal{E}_{\Phi}(\mathbf{x})}\left[\ell_{\mathcal{D}}(\mathbf{x}, \hat{\mathbf{z}}, +1)\right] + \mathbb{E}_{\mathbf{z}\sim P_{\mathbf{z}}, \hat{\mathbf{x}}\sim\mathcal{G}_{\Phi}(\mathbf{z})}\left[\ell_{\mathcal{D}}(\hat{\mathbf{x}}, \mathbf{z}, -1)\right]
\end{aligned}
$$

where $h(t) = \max(0, 1 - t)$ is a "hinge" used to regularize the discriminator [22, 30] [3], also used in BigGAN [1]. The discriminator $\mathcal{D}$ includes three submodules: $F$, $H$, and $J$. $F$ takes only $\mathbf{x}$ as input and $H$ takes only $\mathbf{z}$, and learned projections of their outputs with parameters $\theta_{\mathbf{x}}$ and $\theta_{\mathbf{z}}$ respectively give the scalar unary scores $s_{\mathbf{x}}$ and $s_{\mathbf{z}}$. In our experiments, the data $\mathbf{x}$ are images and latents $\mathbf{z}$ are unstructured flat vectors; accordingly, $F$ is a ConvNet and $H$ is an MLP. The joint score $s_{\mathbf{xz}}$ tying $\mathbf{x}$ and $\mathbf{z}$ is given by the remaining $\mathcal{D}$ submodule, $J$, a function of the outputs of $F$ and $H$.

The $\mathcal{E}$ and $\mathcal{G}$ parameters $\Phi$ are optimized to minimize the loss $\mathcal{L}_{\mathcal{E}\mathcal{G}}$, and the $\mathcal{D}$ parameters $\Theta$ are optimized to minimize loss $\mathcal{L}_{\mathcal{D}}$. As usual, the expectations $\mathbb{E}$ are estimated by Monte Carlo samples taken over minibatches.

Like in BiGAN [5] and ALI [8], the discriminator loss $\mathcal{L}_{\mathcal{D}}$ intuitively trains the discriminator to distinguish between the two joint data-latent distributions from the encoder and the generator, pushing it to predict positive values for encoder input pairs $(\mathbf{x}, \mathcal{E}(\mathbf{x}))$ and negative values for generator input pairs $(\mathcal{G}(\mathbf{z}), \mathbf{z})$. The generator and encoder loss $\mathcal{L}_{\mathcal{E}\mathcal{G}}$ trains these two modules to fool the discriminator into incorrectly predicting the opposite, in effect pushing them to create matching joint data-latent distributions. (In the case of deterministic $\mathcal{E}$ and $\mathcal{G}$, this requires the two modules to invert one another [5].)

## 3 Evaluation

Most of our experiments follow the standard protocol used to evaluate unsupervised learning techniques, first proposed in [34]. We train a BigBiGAN on unlabeled ImageNet, freeze its learned representation, and then train a linear classifier on its outputs, fully supervised using all of the training set labels. We also measure image generation performance, reporting Inception Score [28] (IS) and Fréchet Inception Distance [15] (FID) as the standard metrics there.

### 3.1 Ablation

We begin with an extensive ablation study in which we directly evaluate a number of modeling choices, with results presented in Table 1. Where possible we performed three runs of each variant with different seeds and report the mean and standard deviation for each metric.

We start with a relatively fully-fledged version of the model at $128 \times 128$ resolution (row *Base*), with the $\mathcal{G}$ architecture and the $F$ component of $\mathcal{D}$ taken from the corresponding $128 \times 128$ architectures in BigGAN, including the skip connections and shared noise embedding proposed in [1]. $\mathbf{z}$ is 120 dimensions, split into six groups of 20 dimensions fed into each of the six layers of $\mathcal{G}$ as in [1]. The remaining components of $\mathcal{D} - H$ and $J$ – are 8-layer MLPs with ResNet-style skip connections (four residual blocks with two layers each) and size 2048 hidden layers. The $\mathcal{E}$ architecture is the ResNet-v2-50 ConvNet originally proposed for image classification in [13], followed by a 4-layer MLP (size 4096) with skip connections (two residual blocks) after ResNet's globally average pooled output. The unconditional BigGAN training setup corresponds to the "Single Label" setup proposed in [23], where a single "dummy" label is used for all images (theoretically equivalent to learning a bias in place of the class-conditional batch norm inputs). We then ablate several aspects of the model, with results detailed in the following paragraphs. Additional architectural and optimization details are provided in Appendix A (supplementary material). Full learning curves for many results are included in Appendix D (supplementary material).

**Latent distribution $P_{\mathbf{z}}$ and stochastic $\mathcal{E}$.** As in ALI [8], the encoder $\mathcal{E}$ of our *Base* model is non-deterministic, parametrizing a distribution $\mathcal{N}(\mu, \sigma)$. $\mu$ and $\hat{\sigma}$ are given by a linear layer at the output of the model, and the final standard deviation $\sigma$ is computed from $\hat{\sigma}$ using a non-negative "softplus" non-linearity $\sigma = \log(1 + \exp(\hat{\sigma}))$ [7]. The final $\mathbf{z}$ uses the reparametrized sampling from [19], with $\mathbf{z} = \mu + \epsilon\sigma$, where $\epsilon \sim \mathcal{N}(0, I)$. Compared to a deterministic encoder (row *Deterministic $\mathcal{E}$*) which predicts $\mathbf{z}$ directly without sampling (effectively modeling $P(\mathbf{z}|\mathbf{x})$ as a Dirac $\delta$ distribution), the non-deterministic *Base* model achieves significantly better classification performance (at no cost to generation). We also compared to using a uniform $P_{\mathbf{z}} = \mathcal{U}(-1, 1)$ (row *Uniform $P_{\mathbf{z}}$*) with $\mathcal{E}$ deterministically predicting $\mathbf{z} = \tanh(\hat{\mathbf{z}})$ given a linear output $\hat{\mathbf{z}}$, as done in BiGAN [5]. This also achieves worse classification results than the non-deterministic *Base* model.

**Unary loss terms.** We evaluate the effect of removing one or both unary terms of the loss function proposed in Section 2, $s_{\mathbf{x}}$ and $s_{\mathbf{z}}$. Removing both unary terms (row *No Unaries*) corresponds to the original objective proposed in [5, 8]. It is clear that the $\mathbf{x}$ unary term has a large positive effect on generation performance, with the *Base* and $\mathbf{x}$ *Unary Only* rows having significantly better IS and FID than the $\mathbf{z}$ *Unary Only* and *No Unaries* rows. This result makes intuitive sense as it matches the standard generator loss. It also marginally improves classification performance. The $\mathbf{z}$ unary term makes a more marginal difference, likely due to the relative ease of modeling relatively simple

distributions like isotropic Gaussians, though also does result in slightly improved classification and generation in terms of FID – especially without the **x** term (**z** *Unary Only* vs. *No Unaries*). On the other hand, IS is worse with the **z** term. This may be due to IS roughly measuring the generator's coverage of the major modes of the distribution (the classes) rather than the distribution in its entirety, the latter of which may be better captured by FID and more likely to be promoted by a good encoder $\mathcal{E}$. The requirement of invertibility in a (Big)BiGAN could be encouraging the generator to produce distinguishable outputs across the entire latent space, rather than "collapsing" large volumes of latent space to a single mode of the data distribution.

$\mathcal{G}$ **capacity.** To address the question of the importance of the generator $\mathcal{G}$ in representation learning, we vary the capacity of $\mathcal{G}$ (with $\mathcal{E}$ and $\mathcal{D}$ fixed) in the *Small $\mathcal{G}$* rows. With a third of the capacity of the *Base $\mathcal{G}$* model (*Small $\mathcal{G}$ (32)*), the overall model is quite unstable and achieves significantly worse classification results than the higher capacity base model[4] With two-thirds capacity (*Small $\mathcal{G}$ (64)*), generation performance is substantially worse (matching the results in [1]) and classification performance is modestly worse. These results confirm that a powerful image generator is indeed important for learning good representations via the encoder. Assuming this relationship holds in the future, we expect that better generative models are likely to lead to further improvements in representation learning.

**Standard GAN.** We also compare BigBiGAN's image generation performance against a standard unconditional BigGAN with no encoder $\mathcal{E}$ and only the standard $F$ ConvNet in the discriminator, with only the $s_{\mathbf{x}}$ term in the loss (row *No $\mathcal{E}$ (GAN)*). While the standard GAN achieves a marginally better IS, the BigBiGAN FID is about the same, indicating that the addition of the BigBiGAN $\mathcal{E}$ and joint $\mathcal{D}$ does not compromise generation with the newly proposed unary loss terms described in Section 2. (In comparison, the versions of the model without unary loss term on **x** – rows **z** *Unary Only* and *No Unaries* – have substantially worse generation performance in terms of FID than the standard GAN.) We conjecture that the IS is worse for similar reasons that the $s_{\mathbf{z}}$ unary loss term leads to worse IS. Next we will show that with an enhanced $\mathcal{E}$ taking higher input resolutions, generation with BigBiGAN in terms of FID is substantially improved over the standard GAN.

**High resolution $\mathcal{E}$ with varying resolution $\mathcal{G}$.** BiGAN [5] proposed an asymmetric setup in which $\mathcal{E}$ takes higher resolution images than $\mathcal{G}$ outputs and $\mathcal{D}$ takes as input, showing that an $\mathcal{E}$ taking $128 \times 128$ inputs with a $64 \times 64$ $\mathcal{G}$ outperforms a $64 \times 64$ $\mathcal{E}$ for downstream tasks. We experiment with this setup in BigBiGAN, raising the $\mathcal{E}$ input resolution to $256 \times 256$ – matching the resolution used in typical supervised ImageNet classification setups – and varying the $\mathcal{G}$ output and $\mathcal{D}$ input resolution in $\{64, 128, 256\}$. Our results in Table 1 (rows *High Res $\mathcal{E}$ (256)* and *Low/High Res $\mathcal{G}$ (*)*) show that BigBiGAN achieves better representation learning results as the $\mathcal{G}$ resolution increases, up to the full $\mathcal{E}$ resolution of $256 \times 256$. However, because the overall model is much slower to train with $\mathcal{G}$ at $256 \times 256$ resolution, the remainder of our results use the $128 \times 128$ resolution for $\mathcal{G}$.

Interestingly, with the higher resolution $\mathcal{E}$, generation improves significantly (especially by FID), despite $\mathcal{G}$ operating at the same resolution (row *High Res $\mathcal{E}$ (256)* vs. *Base*). This is an encouraging result for the potential of BigBiGAN as a means of improving adversarial image synthesis itself, besides its use in representation learning and inference.

$\mathcal{E}$ **architecture.** Keeping the $\mathcal{E}$ input resolution fixed at 256, we experiment with varied and often larger $\mathcal{E}$ architectures, including several of the ResNet-50 variants explored in [20]. In particular, we expand the capacity of the hidden layers by a factor of 2 or 4, as well as swap the residual block structure to a reversible variant called *RevNet* [10] with the same number of layers and capacity as the corresponding ResNets. (We use the version of RevNet described in [20].) We find that the base ResNet-50 model (row *High Res $\mathcal{E}$ (256)*) outperforms RevNet-50 (row *RevNet*), but as the network widths are expanded, we begin to see improvements from RevNet-50, with double-width RevNet outperforming a ResNet of the same capacity (rows *RevNet* $\times 2$ and *ResNet* $\times 2$). We see further gains with an even larger quadruple-width RevNet model (row *RevNet* $\times 4$), which we use for our final results in Section 3.2.

| | Encoder ($\mathcal{E}$) | | | | | | Gen. ($\mathcal{G}$) | | Loss $\mathcal{L}_*$ | | | | Results | | |
|---|---|---|---|---|---|---|---|---|---|---|---|---|---|---|---|
| | A. | D. | C. | R. | Var. | $\eta$ | C. | R. | $s_{\mathbf{xz}}$ | $s_{\mathbf{x}}$ | $s_{\mathbf{z}}$ | $P_{\mathbf{z}}$ | IS ($\uparrow$) | FID ($\downarrow$) | Cls. ($\uparrow$) |
| Base | S | 50 | 1 | 128 | ✓ | 1 | 96 | 128 | ✓ | ✓ | ✓ | $\mathcal{N}$ | $22.66 \pm 0.18$ | $31.19 \pm 0.37$ | $48.10 \pm 0.13$ |
| Deterministic $\mathcal{E}$ | S | 50 | 1 | 128 | (-) | 1 | 96 | 128 | ✓ | ✓ | ✓ | $\mathcal{N}$ | $22.79 \pm 0.27$ | $31.31 \pm 0.30$ | $46.97 \pm 0.35$ |
| Uniform $P_{\mathbf{z}}$ | S | 50 | 1 | 128 | (-) | 1 | 96 | 128 | ✓ | ✓ | ✓ | $(\mathcal{U})$ | $22.83 \pm 0.24$ | $31.52 \pm 0.28$ | $45.11 \pm 0.93$ |
| $\mathbf{x}$ Unary Only | S | 50 | 1 | 128 | ✓ | 1 | 96 | 128 | ✓ | ✓ | (-) | $\mathcal{N}$ | $23.19 \pm 0.28$ | $31.99 \pm 0.30$ | $47.74 \pm 0.20$ |
| $\mathbf{z}$ Unary Only | S | 50 | 1 | 128 | ✓ | 1 | 96 | 128 | ✓ | (-) | ✓ | $\mathcal{N}$ | $19.52 \pm 0.39$ | $39.48 \pm 1.00$ | $47.78 \pm 0.28$ |
| No Unaries (BiGAN) | S | 50 | 1 | 128 | ✓ | 1 | 96 | 128 | ✓ | (-) | (-) | $\mathcal{N}$ | $19.70 \pm 0.30$ | $42.92 \pm 0.92$ | $46.71 \pm 0.88$ |
| Small $\mathcal{G}$ (32) | S | 50 | 1 | 128 | ✓ | 1 | (32) | 128 | ✓ | ✓ | ✓ | $\mathcal{N}$ | $3.28 \pm 0.18$ | $247.30 \pm 10.31$ | $43.59 \pm 0.34$ |
| Small $\mathcal{G}$ (64) | S | 50 | 1 | 128 | ✓ | 1 | (64) | 128 | ✓ | ✓ | ✓ | $\mathcal{N}$ | $19.96 \pm 0.15$ | $38.93 \pm 0.39$ | $47.54 \pm 0.33$ |
| No $\mathcal{E}$ (GAN) * | | | (-) | | | | 96 | 128 | (-) | ✓ | (-) | $\mathcal{N}$ | $23.56 \pm 0.37$ | $30.91 \pm 0.23$ | - |
| High Res $\mathcal{E}$ (256) | S | 50 | 1 | (256) | ✓ | 1 | 96 | 128 | ✓ | ✓ | ✓ | $\mathcal{N}$ | $23.45 \pm 0.14$ | $27.86 \pm 0.13$ | $50.80 \pm 0.30$ |
| Low Res $\mathcal{G}$ (64) | S | 50 | 1 | (256) | ✓ | 1 | 96 | (64) | ✓ | ✓ | ✓ | $\mathcal{N}$ | $19.40 \pm 0.19$ | $15.82 \pm 0.06$ | $47.51 \pm 0.09$ |
| High Res $\mathcal{G}$ (256) | S | 50 | 1 | (256) | ✓ | 1 | 96 | (256) | ✓ | ✓ | ✓ | $\mathcal{N}$ | 24.70 | 38.58 | 51.49 |
| ResNet-101 | S | (101) | 1 | (256) | ✓ | 1 | 96 | 128 | ✓ | ✓ | ✓ | $\mathcal{N}$ | 23.29 | 28.01 | 51.21 |
| ResNet $\times 2$ | S | 50 | (2) | (256) | ✓ | 1 | 96 | 128 | ✓ | ✓ | ✓ | $\mathcal{N}$ | 23.68 | 27.81 | 52.66 |
| RevNet | (V) | 50 | 1 | (256) | ✓ | 1 | 96 | 128 | ✓ | ✓ | ✓ | $\mathcal{N}$ | $23.33 \pm 0.09$ | $27.78 \pm 0.06$ | $49.42 \pm 0.18$ |
| RevNet $\times 2$ | (V) | 50 | (2) | (256) | ✓ | 1 | 96 | 128 | ✓ | ✓ | ✓ | $\mathcal{N}$ | 23.21 | 27.96 | 54.40 |
| RevNet $\times 4$ | (V) | 50 | (4) | (256) | ✓ | 1 | 96 | 128 | ✓ | ✓ | ✓ | $\mathcal{N}$ | 23.23 | 28.15 | 57.15 |
| ResNet ($\uparrow \mathcal{E}$ LR) | S | 50 | 1 | (256) | ✓ | (10) | 96 | 128 | ✓ | ✓ | ✓ | $\mathcal{N}$ | $23.27 \pm 0.22$ | $28.51 \pm 0.44$ | $53.70 \pm 0.15$ |
| RevNet $\times 4$ ($\uparrow \mathcal{E}$ LR) | (V) | 50 | (4) | (256) | ✓ | (10) | 96 | 128 | ✓ | ✓ | ✓ | $\mathcal{N}$ | 23.08 | 28.54 | 60.15 |

Table 1: Results for variants of BigBiGAN, given in Inception Score [28] (IS) and Fréchet Inception Distance [15] (FID) of the generated images, and ImageNet top-1 classification accuracy percentage (Cls.) of a supervised logistic regression classifier trained on the encoder features [34], computed on a split of 10K images randomly sampled from the training set, which we refer to as the "train$_{\text{val}}$" split. The *Encoder ($\mathcal{E}$)* columns specify the $\mathcal{E}$ architecture (A.) as ResNet (S) or RevNet (V), the depth (D., e.g. 50 for ResNet-50), the channel width multiplier (C.), with 1 denoting the original widths from [13], the input image resolution (R.), whether the variance is predicted and a $\mathbf{z}$ vector is sampled from the resulting distribution (Var.), and the learning rate multiplier $\eta$ relative to the $\mathcal{G}$ learning rate. The *Generator ($\mathcal{G}$)* columns specify the BigGAN $\mathcal{G}$ channel multiplier (C.), with 96 corresponding to the original width from [1], and output image resolution (R.). The *Loss* columns specify which terms of the BigBiGAN loss are present in the objective. The $P_z$ column specifies the input distribution as a standard normal $\mathcal{N}(0,1)$ or continuous uniform $\mathcal{U}(-1,1)$. Changes from the *Base* setup in each row are highlighted in blue. Results with margins of error (written as "$\mu \pm \sigma$") are the means and standard deviations over three runs with different random seeds. (Experiments requiring more computation were run only once.) (* Result for vanilla GAN (*No $\mathcal{E}$ (GAN)*) selected with early stopping based on best FID; other results selected with early stopping based on validation classification accuracy (Cls.).)

**Decoupled $\mathcal{E}$/$\mathcal{G}$ optimization.** As a final improvement, we decoupled the $\mathcal{E}$ optimizer from that of $\mathcal{G}$, and found that simply using a $10\times$ higher learning rate for $\mathcal{E}$ dramatically accelerates training and improves final representation learning results. For ResNet-50 this improves linear classifier accuracy by nearly 3% (*ResNet ($\uparrow \mathcal{E}$ LR)* vs. *High Res $\mathcal{E}$ (256)*). We also applied this to our largest $\mathcal{E}$ architecture, RevNet-50 $\times 4$, and saw similar gains (*RevNet $\times 4$ ($\uparrow \mathcal{E}$ LR)* vs. *RevNet $\times 4$*).

## 3.2 Comparison with prior methods

**Representation learning.** We now take our best model by train$_{\text{val}}$ classification accuracy from the above ablations and present results on the official ImageNet validation set, comparing against the state of the art in recent unsupervised learning literature. For comparison, we also present classification results for our best performing variant with the smaller ResNet-50-based $\mathcal{E}$. These models correspond to the last two rows of Table 1, *ResNet ($\uparrow \mathcal{E}$ LR)* and *RevNet $\times 4$ ($\uparrow \mathcal{E}$ LR)*.

Results are presented in Table 2. (For reference, the fully supervised accuracy of these architectures is given in Appendix A, Table 1 (supplementary material).) Compared with a number of modern self-supervised approaches [25, 3, 34, 32, 9, 14] and combinations thereof [4], our BigBiGAN approach based purely on generative models performs well for representation learning, state-of-the-art among recent unsupervised learning results, improving upon a recently published result from [20] of 55.4% to 60.8% top-1 accuracy using rotation prediction pre-training with the same representation learning

| Method | Architecture | Feature | Top-1 | Top-5 |
|---|---|---|---|---|
| BiGAN [5, 35] | AlexNet | Conv3 | 31.0 | - |
| SS-GAN [2] | ResNet-19 | Block6 | 38.3 | - |
| Motion Segmentation (MS) [25, 4] | ResNet-101 | AvePool | 27.6 | 48.3 |
| Exemplar (Ex) [6, 4] | ResNet-101 | AvePool | 31.5 | 53.1 |
| Relative Position (RP) [3, 4] | ResNet-101 | AvePool | 36.2 | 59.2 |
| Colorization (Col) [34, 4] | ResNet-101 | AvePool | 39.6 | 62.5 |
| Combination of MS+Ex+RP+Col [4] | ResNet-101 | AvePool | - | 69.3 |
| CPC [32] | ResNet-101 | AvePool | 48.7 | 73.6 |
| Rotation [9, 20] | RevNet-50 $\times 4$ | AvePool | 55.4 | - |
| Efficient CPC [14] | ResNet-170 | AvePool | 61.0 | 83.0 |
| BigBiGAN (ours) | ResNet-50 | AvePool | 55.4 | 77.4 |
| | ResNet-50 | BN+CReLU | 56.6 | 78.6 |
| | RevNet-50 $\times 4$ | AvePool | 60.8 | 81.4 |
| | RevNet-50 $\times 4$ | BN+CReLU | 61.3 | 81.9 |

Table 2: Comparison of BigBiGAN models on the official ImageNet validation set against recent competing approaches with a supervised logistic regression classifier. BigBiGAN results are selected with early stopping based on highest accuracy on our train$_{val}$ subset of 10K training set images. *ResNet-50* results correspond to row *ResNet ($\uparrow \mathcal{E}$ LR)* in Table 1, and *RevNet-50* $\times 4$ corresponds to *RevNet* $\times 4$ *($\uparrow \mathcal{E}$ LR)*.

| Method | Steps | IS ($\uparrow$) | FID vs. Train ($\downarrow$) | FID vs. Val. ($\downarrow$) |
|---|---|---|---|---|
| BigGAN + SL [23] | 500K | 20.4 (15.4 $\pm$ 7.57) | - | 25.3 (71.7 $\pm$ 66.32) |
| BigGAN + Clustering [23] | 500K | 22.7 (22.8 $\pm$ 0.42) | - | 23.2 (22.7 $\pm$ 0.80) |
| BigBiGAN + SL (ours) | 500K | 25.38 (25.33 $\pm$ 0.17) | 22.78 (22.63 $\pm$ 0.23) | 23.60 (23.56 $\pm$ 0.12) |
| BigBiGAN High Res $\mathcal{E}$ + SL (ours) | 500K | 25.43 (25.45 $\pm$ 0.04) | 22.34 (22.36 $\pm$ 0.04) | 22.94 (23.00 $\pm$ 0.15) |
| BigBiGAN High Res $\mathcal{E}$ + SL (ours) | 1M | 27.94 (27.80 $\pm$ 0.21) | 20.32 (20.27 $\pm$ 0.09) | 21.61 (21.62 $\pm$ 0.09) |

Table 3: Comparison of our BigBiGAN for unsupervised (unconditional) generation vs. previously reported results for unsupervised BigGAN from [23]. We specify the "pseudo-labeling" method as *SL* (Single Label) or *Clustering*. For comparison we train BigBiGAN for the same number of steps (500K) as the BigGAN-based approaches from [23], but also present results from additional training to 1M steps in the last row and observe further improvements. All results above include the median $m$ as well as the mean $\mu$ and standard deviation $\sigma$ across three runs, written as "$m$ ($\mu \pm \sigma$)". The BigBiGAN result is selected with early stopping based on best FID vs. Train.

architecture[5] and feature, labeled as *AvePool* in Table 2, and matches the results of the concurrent work in [14] based on contrastive predictive coding (CPC).

We also experiment with learning linear classifiers on a different rendering of the *AvePool* feature, labeled *BN+CReLU*, which boosts our best results with RevNet $\times 4$ to 61.3% top-1 accuracy. Given the global average pooling output $a$, we first compute $h = \text{BatchNorm}(a)$, and the final feature is computed by concatenating $[\text{ReLU}(h), \text{ReLU}(-h)]$, sometimes called a "CReLU" (concatenated ReLU) non-linearity [29]. $\text{BatchNorm}$ denotes parameter-free Batch Normalization [16], where the scale ($\gamma$) and offset ($\beta$) parameters are not learned, so training a linear classifier on this feature does not involve any additional learning. The CReLU non-linearity retains all the information in its inputs and doubles the feature dimension, each of which likely contributes to the improved results.

Finally, in Appendix C (supplementary material) we consider evaluating representations by zero-shot $k$ nearest neighbors classification, achieving 43.3% top-1 accuracy in this setting. Qualitative examples of nearest neighbors are presented in Appendix C, Figure 11 (supplementary material).

**Unsupervised image generation.** In Table 3 we show results for unsupervised generation with BigBiGAN, comparing to the BigGAN-based [1] unsupervised generation results from [23]. Note

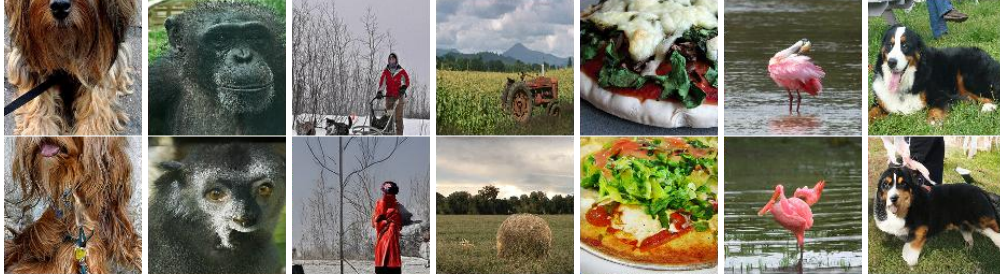

Figure 2: Selected reconstructions from an unsupervised BigBiGAN model (Section 3.3). Top row images are real data $\mathbf{x} \sim P_{\mathbf{x}}$; bottom row images are generated reconstructions of the above image $\mathbf{x}$ computed by $\mathcal{G}(\mathcal{E}(\mathbf{x}))$. Unlike most explicit reconstruction costs (e.g., pixel-wise), the reconstruction cost implicitly minimized by a (Big)BiGAN [5, 8] tends to emphasize more semantic, high-level details. Additional reconstructions are presented in Appendix B (supplementary material).

that these results differ from those in Table 1 due to the use of the data augmentation method of [23][6] (rather than ResNet-style preprocessing used for all results in our Table 1 ablation study). The lighter augmentation from [23] results in better image generation performance under the IS and FID metrics. The improvements are likely due in part to the fact that this augmentation, on average, crops larger portions of the image, thus yielding generators that typically produce images encompassing most or all of a given object, which tends to result in more representative samples of any given class (giving better IS) and more closely matching the statistics of full center crops (as used in the real data statistics to compute FID). Besides this preprocessing difference, the approaches in Table 3 have the same configurations as used in the *Base* or *High Res $\mathcal{E}$ (256)* row of Table 1.

These results show that BigBiGAN significantly improves both IS and FID over the baseline unconditional BigGAN generation results with the same (unsupervised) "labels" (a single fixed label in the *SL* (Single Label) approach – row *BigBiGAN + SL* vs. *BigGAN + SL*). We see further improvements using a high resolution $\mathcal{E}$ (row *BigBiGAN High Res $\mathcal{E}$ + SL*), surpassing the previous unsupervised state of the art (row *BigGAN + Clustering*) under both IS and FID. (Note that the image generation results remain comparable: the generated image resolution is still $128 \times 128$ here, despite the higher resolution $\mathcal{E}$ input.) The alternative "pseudo-labeling" approach from [23], *Clustering*, which uses labels derived from unsupervised clustering, is complementary to BigBiGAN and combining both could yield further improvements. Finally, observing that results continue to improve significantly with training beyond 500K steps, we also report results at 1M steps in the final row of Table 3.

## 3.3 Reconstruction

As shown in [5, 8], the (Big)BiGAN $\mathcal{E}$ and $\mathcal{G}$ can reconstruct data instances $\mathbf{x}$ by computing the encoder's predicted latent representation $\mathcal{E}(\mathbf{x})$ and then passing this predicted latent back through the generator to obtain the reconstruction $\mathcal{G}(\mathcal{E}(\mathbf{x}))$. We present BigBiGAN reconstructions in Figure 2. These reconstructions are far from pixel-perfect, likely due in part to the fact that no reconstruction cost is explicitly enforced by the objective – reconstructions are not even computed at training time. However, they may provide some intuition for what features the encoder $\mathcal{E}$ learns to model. For example, when the input image contains a dog, person, or a food item, the reconstruction is often a different instance of the same "category" with similar pose, position, and texture – for example, a similar species of dog facing the same direction. The extent to which these reconstructions tend to retain the high-level semantics of the inputs rather than the low-level details suggests that BigBiGAN training encourages the encoder to model the former more so than the latter. Additional reconstructions are presented in Appendix B (supplementary material).

# 4 Related work

A number of approaches to unsupervised representation learning from images based on self-supervision have proven very successful. Self-supervision generally involves learning from tasks designed to resemble supervised learning in some way, but in which the "labels" can be created automatically from the data itself with no manual effort. An early example is relative location prediction [3], where a model is trained on input pairs of image patches and predicts their relative locations. Contrastive predictive coding (CPC) [32, 14] is a recent related approach where, given an image patch, a model predicts which patches occur in other image locations. Other approaches include colorization [34, 35], motion segmentation [25], rotation prediction [9, 2], GAN-based discrimination [26, 2], and exemplar matching [6]. Rigorous empirical comparisons of many of these approaches have also been conducted [4, 20]. A key advantage offered by BigBiGAN and other approaches based on generative models, relative to most self-supervised approaches, is that their input may be the full-resolution image or other signal, with no cropping or modification of the data needed (though such modifications may be beneficial as data augmentation). This means the resulting representation can typically be applied directly to full data in the downstream task with no domain shift.

A number of relevant autoencoder and GAN variants have also been proposed. Associative compression networks (ACNs) [12] learn to compress at the dataset level by conditioning data on other previously transmitted data which are similar in code space, resulting in models that can "daydream" semantically similar samples, similar to BigBiGAN reconstructions. VQ-VAEs [33] pair a discrete (vector quantized) encoder with an autoregressive decoder to produce faithful reconstructions with a high compression factor and demonstrate representation learning results in reinforcement learning settings. In the adversarial space, adversarial autoencoders [24] proposed an autoencoder-style encoder-decoder pair trained with pixel-level reconstruction cost, replacing the KL-divergence regularization of the prior used in VAEs [19] with a discriminator. In another proposed VAE-GAN hybrid [21] the pixel-space reconstruction error used in most VAEs is replaced with feature space distance from an intermediate layer of a GAN discriminator. Other hybrid approaches like AGE [31] and $\alpha$-GAN [27] add an encoder to stabilize GAN training. An interesting difference between many of these approaches and the BiGAN [8, 5] framework is that BiGAN does not train the encoder or generator with an explicit reconstruction cost. Though it can be shown that (Big)BiGAN implicitly minimizes a reconstruction cost, qualitative reconstruction results (Section 3.3) suggest that this reconstruction cost is of a different flavor, emphasizing high-level semantics over pixel-level details.

# 5 Discussion

We have shown that BigBiGAN, an unsupervised learning approach based purely on generative models, achieves state-of-the-art results in image representation learning on ImageNet. Our ablation study lends further credence to the hope that powerful generative models can be beneficial for representation learning, and in turn that learning an inference model can improve large-scale generative models. In the future we hope that representation learning can continue to benefit from further advances in generative models and inference models alike, as well as scaling to larger image databases.

### Acknowledgments

The authors would like to thank Aidan Clark, Olivier Hénaff, Aäron van den Oord, Sander Dieleman, and many other colleagues at DeepMind for useful discussions and feedback on this work.

## Footnotes

[1] Models available at https://tfhub.dev/s?publisher=deepmind&q=bigbigan, with a Colab notebook demo at https://colab.research.google.com/github/tensorflow/hub/blob/master/examples/colab/bigbigan_with_tf_hub.ipynb.

[2]See footnote [1].

[3] We also considered an alternative discriminator loss $\ell_{\mathcal{D}}'$ which invokes the "hinge" $h$ just once on the sum of the three loss terms $-\ell_{\mathcal{D}}'(\mathbf{x}, \mathbf{z}, y) = h(y\left(s_{\mathbf{x}}(\mathbf{x}) + s_{\mathbf{z}}(\mathbf{z}) + s_{\mathbf{xz}}(\mathbf{x}, \mathbf{z})\right))$ – but found that this performed significantly worse than $\ell_{\mathcal{D}}$ above which clamps each of the three loss terms separately.

[4]Though the generation performance by IS and FID in row *Small $\mathcal{G}$ (32)* is very poor at the point we measured – when its best validation classification performance (43.59%) is achieved – this model was performing more reasonably for generation earlier in training, reaching IS 14.69 and FID 60.67.

[5]Our RevNet $\times 4$ architecture matches the widest architectures used in [20], labeled as $\times 16$ there.

[6]See the "distorted" preprocessing method from the Compare GAN framework: https://github.com/google/compare_gan/blob/master/compare_gan/datasets.py.

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
