[Supplementary Material]

# Supplementary Material for:
# Large Scale Adversarial Representation Learning

**Jeff Donahue**
DeepMind
jeffdonahue@google.com

**Karen Simonyan**
DeepMind
simonyan@google.com

## Appendix A   Model and optimization details

Our optimizer matches that of BigGAN [2] – we use Adam [7] with batch size 2048 and the same learning rates and other hyperparameters, using the $\mathcal{G}$ optimizer to update $\mathcal{E}$ simultaneously, with the same alternating optimization: two $\mathcal{D}$ updates followed by a single joint update of $\mathcal{G}$ and $\mathcal{E}$. (We do not use orthogonal regularization used in [2], finding it gave worse results in the unconditional setting, matching the findings of [8].) Spectral normalization [9] is used in $\mathcal{G}$ and $\mathcal{D}$, but not in $\mathcal{E}$. Full cross-replica batch normalization is used in both $\mathcal{G}$ and $\mathcal{E}$ (including for the linear classifier training on $\mathcal{E}$ features used for evaluations). We also apply exponential moving averaging (EMA) with a decay of 0.9999 to the $\mathcal{G}$ and $\mathcal{E}$ weights in all evaluations. (We find this results in only a small improvement for $\mathcal{E}$ evaluations, but a substantial one for $\mathcal{G}$ evaluations.)

At BigBiGAN training time, as well as linear classification evaluation training time, we preprocess inputs with ResNet [6]-style data augmentation, though with crops of size 128 or 256 rather than 224[1].

For linear classification evaluations in the ablations reported in Table 1 (main text), we hold out 10K randomly selected images from the official ImageNet [11] training set as a validation set and report accuracy on that validation set, which we call $train_{val}$. All results in Table 1 (main text) are run for 500K steps, with early stopping based on linear classifier accuracy on our $train_{val}$ split. In all of these models the linear classifier is initialized to 0 and trained for 5K Adam steps with a (high) learning rate of 0.01 and EMA smoothing with decay 0.9999. We have found it helpful to monitor representation learning progress during BigBiGAN training by periodically rerunning this linear classification evaluation from scratch given the current $\mathcal{E}$ weights, resetting the classifier weights to 0 before each evaluation.

In Table 2 (main text) we extend the BigBiGAN training time to 1M steps, and report results on the official validation set of 50K images for comparison with prior work. The classifier in these results is trained for 100K Adam steps, sweeping over learning rates $\{10^{-4}, 3 \cdot 10^{-4}, 10^{-3}, 3 \cdot 10^{-3}, 10^{-2}\}$, again applying EMA with decay 0.9999 to the classifier weights. Hyperparameter selection and early stopping is again based on classification accuracy on $train_{val}$. As in [2], FID is reported against statistics over the full ImageNet training set, preprocessed by resizing the minor axis to the $\mathcal{G}$ output resolution and taking the center crop along the major axis, except as noted in Table 3 (main text), where we also report FID against the validation set for comparison with [8].

All models were trained via TensorFlow [1] and Sonnet [10] with data parallelism on TPU pod slices [5] using 32 to 512 cores, coordinated by TF-Replicator [3].

**Supervised model performance.**   In Table 1 we present the results of fully supervised training with the model architectures used in our experiments in Section 3 (main text) for comparison purposes.

| Architecture | Top-1 | Top-5 |
|---|---|---|
| ResNet-50 | 76.3 | 93.1 |
| ResNet-101 | 77.8 | 93.8 |
| RevNet-50 | 71.8 | 90.5 |
| RevNet-50 $\times 2$ | 74.9 | 92.2 |
| RevNet-50 $\times 4$ | 76.6 | 93.1 |

Table 1: ImageNet validation set accuracy for fully supervised end-to-end training of the model architectures used in our representation learning experiments.

(a) RevNet $\times 4$        (b) $+ \uparrow \mathcal{E}$ LR

Figure 1: Visualization of first layer convolutional filters for our unsupervised BigBiGAN models with the RevNet $\times 4$ $\mathcal{E}$ architecture, which includes 1024 filters. (Best viewed with zoom.)

**First layer convolutional filters.** In Figure 1 we visualize the learned convolutional filters for the first convolutional layer of our BigBiGAN encoders $\mathcal{E}$ using the largest RevNet $\times 4$ $\mathcal{E}$ architecture. Note the difference between the filters in (a) and (b) (corresponding to rows *RevNet* $\times 4$ and *RevNet* $\times 4$ *($\uparrow \mathcal{E}$ LR)* in Table 1 (main text)). In (b) we use the higher $\mathcal{E}$ learning rate and see a corresponding qualitative improvement in the appearance of the learned filters, with less noise and more Gabor-like and color filters, as observed in BiGAN [4]. This suggests that examining the convolutional filters of the input layer can serve as a diagnostic for undertrained models.

| Model | Samples | | | Reconstructions | |
|---|---|---|---|---|---|
| | Image | IS ($\uparrow$) | FID ($\downarrow$) | Image | Rel. $\ell_1$ Error % ($\downarrow$) |
| Base | Figure 2 | 24.10 | 30.14 | Figure 3 | 70.54 |
| Light Augmentation | Figure 4 | 27.09 | 20.96 | Figure 5 | 72.53 |
| High Res $\mathcal{E}$ (256) | Figure 6 | 24.91 | 26.56 | Figure 7 | 70.60 |
| High Res $\mathcal{G}$ (256) | Figure 8 | 25.73 | 37.21 | Figure 9 | 77.70 |

Table 2: Links to BigBiGAN samples and reconstructions with associated metrics.

## Appendix B    Samples and reconstructions

In this Appendix we present BigBiGAN samples and reconstructions from several variants of the method. Table 2 includes pointers to samples and reconstruction images, as well as relevant metrics. The samples were selected by best FID vs. training set statistics, and we show the IS and FID along with sample images at that point. The reconstructions were selected by best (lowest) relative pixel-wise $\ell_1$ error, the error metric presented in Table 2, computed as:

$$E_{\mathrm{Rel}\ell_1} = \frac{\mathbb{E}_{\mathbf{x} \sim P_{\mathbf{x}}} ||\mathbf{x} - \mathcal{G}(\mathcal{E}(\mathbf{x}))||_1}{\mathbb{E}_{\mathbf{x},\mathbf{x}' \sim P_{\mathbf{x}}} ||\mathbf{x}' - \mathcal{G}(\mathcal{E}(\mathbf{x}))||_1},$$

where $\mathbf{x}$ and $\mathbf{x}'$ are independent data samples, and $||\mathbf{x}' - \mathcal{G}(\mathcal{E}(\mathbf{x}))||_1$ serves as a "baseline" reconstruction error relative to a "random" input. For example, with a random initialization of $\mathcal{G}$ and $\mathcal{E}$, we have $E_{\mathrm{Rel}\ell_1} \approx 1$. This relative metric penalizes degenerate reconstructions, such as the mean image, which would sometimes achieve low absolute reconstruction error despite having no perceptual similarity to the inputs. despite that the resulting images having no perceptual similarity to the inputs. In practice, given $N$ data samples $\mathbf{x}_0, \mathbf{x}_1, \ldots, \mathbf{x}_{N-1}$ (we use $N = 50$K), we estimate the denominator by comparing each sample $\mathbf{x}_i$ with a single neighbor $\mathbf{x}_{(i+1) \bmod N}$, computing:

$$E_{\mathrm{Rel}\ell_1} \approx \frac{\sum_{i=0}^{N-1} ||\mathbf{x}_i - \mathcal{G}(\mathcal{E}(\mathbf{x}_i))||_1}{\sum_{i=0}^{N-1} ||\mathbf{x}_{(i+1) \bmod N} - \mathcal{G}(\mathcal{E}(\mathbf{x}_i))||_1}$$

**Iterated reconstruction**    To further explore the behavior of a BigBiGAN (or any other model capable of approximately reconstructing its input), we can "iterate" the reconstruction operation. In particular, let $R_i(\mathbf{x})$ be defined for non-negative integers $i$ and input images $\mathbf{x}$ as:

$$R_0(\mathbf{x}) = \mathbf{x}$$
$$R_{i+1}(\mathbf{x}) = \mathcal{G}(\mathcal{E}(R_i(\mathbf{x})))$$

In Figure 10 we show the results of up to 500 steps of this process for a few sample images. Qualitatively, the first several steps of this process often appear to retain some semantics of the input image $\mathbf{x}$. After dozens or hundreds of iterations, however, little content from the original input apparently remains intact.

Figure 2: $128 \times 128$ samples $\hat{\mathbf{x}} \sim \mathcal{G}(\mathbf{z})$ from an unsupervised BigBiGAN generator $\mathcal{G}$, trained using the *Base* method from Table 1 (main text).

Figure 3: $128 \times 128$ reconstructions from an unsupervised BigBiGAN model, trained using the *Base* method from Table 1 (main text). The top rows of each pair are real data $\mathbf{x} \sim P_{\mathbf{x}}$, and bottom rows are generated reconstructions computed by $\mathcal{G}(\mathcal{E}(\mathbf{x}))$.

Figure 4: $128 \times 128$ samples $\hat{\mathbf{x}} \sim \mathcal{G}(\mathbf{z})$ from an unsupervised BigBiGAN generator $\mathcal{G}$, trained using the lighter augmentation from [8] with generation results reported in Table 3 (main text).

Figure 5: $128 \times 128$ reconstructions from an unsupervised BigBiGAN model, trained using the lighter augmentation from [8] with generation results reported in Table 3 (main text). The top rows of each pair are real data $\mathbf{x} \sim P_{\mathbf{x}}$, and bottom rows are generated reconstructions computed by $\mathcal{G}(\mathcal{E}(\mathbf{x}))$.

Figure 6: $128 \times 128$ samples $\hat{\mathbf{x}} \sim \mathcal{G}(\mathbf{z})$ from an unsupervised BigBiGAN generator $\mathcal{G}$, trained using the *High Res $\mathcal{E}$ (256)* configuration from Table 1 (main text).

Figure 7: $128 \times 128$ reconstructions of $256 \times 256$ encoder input images from an unsupervised BigBiGAN model, trained using the *High Res $\mathcal{E}$ (256)* configuration from Table 1 (main text). Reconstructions are upsampled from $128 \times 128$ to $256 \times 256$ for visualization. The top rows of each pair are real data $\mathbf{x} \sim P_{\mathbf{x}}$, and bottom rows are generated reconstructions computed by $\mathcal{G}(\mathcal{E}(\mathbf{x}))$.

Figure 8: $256 \times 256$ samples $\hat{\mathbf{x}} \sim \mathcal{G}(\mathbf{z})$ from an unsupervised BigBiGAN generator $\mathcal{G}$, trained with a high-resolution $\mathcal{E}$ and $\mathcal{G}$ (*High Res $\mathcal{G}$ (256)* from Table 1 (main text)).

Figure 9: $256 \times 256$ reconstructions from an unsupervised BigBiGAN model, trained with a high-resolution $\mathcal{E}$ and $\mathcal{G}$ (*High Res $\mathcal{G}$ (256)* from Table 1 (main text)). The top rows of each pair are real data $\mathbf{x} \sim P_{\mathbf{x}}$, and bottom rows are generated reconstructions computed by $\mathcal{G}(\mathcal{E}(\mathbf{x}))$.

| $R_0 : R_9$ | $R_{10} : R_{500}$ | $R_0 : R_9$ | $R_{10} : R_{500}$ | $R_0 : R_9$ | $R_{10} : R_{500}$ |
|---|---|---|---|---|---|
| Image 1 | | Image 2 | | Image 3 | |

Figure 10: Iterated reconstructions from an unsupervised BigBiGAN model, trained using the *ResNet* *(↑ $\mathcal{E}$ LR)* method from Table 1 (main text), computed by recursively running the reconstruction operation $\mathcal{G}(\mathcal{E}(\cdot))$ on its own output as described in Appendix B. In each pair of columns, the left column shows a real input image $R_0$ at the top, and $R_1$ through $R_9$ in the remaining rows, the results of iterating reconstruction one to nine times, The right column shows the result of up to 500 iterations sampled at longer intervals, displaying $R_{10}, R_{20}, R_{30}, R_{40}, R_{50}, R_{100}, R_{200}, R_{300}, R_{400}$, and $R_{500}$.

| Metric | Top-1 / Top-5 Acc. (%) | | | |
|---|---|---|---|---|
| | $k = 1$ | $k = 5$ | $k = 25$ | $k = 50$ |
| $D_1$ | 38.09 / - | 41.28 / 58.56 | 43.32 / 65.12 | 42.73 / 66.22 |
| $D_2$ | 35.68 / - | 38.61 / 55.59 | 40.65 / 62.23 | 40.15 / 63.42 |

Table 3: Accuracy of $k$ nearest neighbors classifiers in BigBiGAN feature space on the ImageNet validation set. We report results under the normalized $\ell_1$ distance $D_1$ as well as the normalized $\ell_2$ (cosine) distance $D_2$.

## Appendix C  Nearest neighbors

In this Appendix we consider an alternative way of evaluating representations — by means of $k$ nearest neighbors classification, which does not involve learning any parameters during evaluation and is even simpler than learning a linear classifier as done in Section 3 (main text). For all results in this section, we use the outputs of the global average pooling layer (a flat 8192D feature) of our best performing model, *RevNet* $\times 4$, $\uparrow \mathcal{E}$ *LR*. We do not do any data augmentation for either the training or validation sets: we simply crop each image at the center of its larger axis and resize to $256 \times 256$.

We use a normalized $\ell_1$ or $\ell_2$ distance metric as our nearest neighbors criterion, defined as $D_p(a, b) = \left\| \frac{a}{||a||_p} - \frac{b}{||b||_p} \right\|_p$, for $p \in \{1, 2\}$. ($D_2$ corresponds to cosine distance.) For label predictions with multiple neighbors ($k > 1$), we use a simple counting scheme: the label with the most votes is selected as the prediction. Ties (multiple labels with the same number of votes) are broken by $k = 1$ nearest neighbor classification among the data with the tied labels.

**Quantitative results.**  In Table 3 we present $k$ nearest neighbors classification results for $k \in \{1, 5, 25, 50\}$. Across all $k$, the $\ell_1$-based metric $D_1$ outperforms $D_2$, and the remainder of our discussion refers to the $D_1$ results. With just a single neighbor ($k = 1$) we achieve a top-1 accuracy around 38%. Top-1 accuracy reaches 43% with $k = 25$, dropping off slightly at $k = 50$ as votes from more distant neighbors are added.

**Qualitative results.**  Figure 11 shows sample nearest neighbors in the ImageNet training set for query images in the validation set. Despite being fully unsupervised, the neighbors in many cases match the query image in terms of high-level semantic content such as the category of the object of interest, demonstrating BigBiGAN's ability to capture high-level attributes of the data in its unsupervised representations. Where applicable, the object's pose and position in the image appears to be important as well – for example, the nearest neighbors of the RV (row 2, column 2) are all RVs facing roughly the same direction. In other cases, the nearest neighbors appear to be selected primarily based on the background or color scheme.

**Discussion.**  While our quantitative $k$ nearest neighbors classification results are far from the state of the art for ImageNet classification and significantly below the linear classifier-based results reported in Table 2 (main text), note that in this setup, no supervised learning of model parameters from labels occurs at any point: labels are predicted purely based on distance in a feature space learned from BigBiGAN training on image pixels alone. We believe this makes nearest neighbors classification an interesting additional benchmark for future approaches to unsupervised representation learning.

Figure 11: Nearest neighbors in BigBiGAN $\mathcal{E}$ feature space, from our best performing model (*RevNet* ×4, ↑ $\mathcal{E}$ *LR*). In each row, the first (left) column is a query image, and the remaining columns are its three nearest neighbors from the training set (the leftmost being the nearest, next being the second nearest, etc.). The query images above are the first 24 images in the ImageNet validation set.

**Inception Score (IS)**

| | |
|---|---|
| 23.99@497500: | No $\mathcal{E}$ (GAN) |
| 23.89@490500: | No $\mathcal{E}$ (GAN) |
| 23.84@487250: | High Res $\mathcal{E}$ |
| 23.62@466000: | High Res $\mathcal{E}$ |
| 23.39@491000: | High Res $\mathcal{E}$ |
| 23.31@495500: | Base |
| 23.31@486500: | No $\mathcal{E}$ (GAN) |
| 23.03@481750: | Base |
| 22.77@465750: | Base |

**Fréchet Inception Distance (FID)**

| | |
|---|---|
| 31.40@496500: | Base |
| 31.22@467000: | No $\mathcal{E}$ (GAN) |
| 30.85@499000: | Base |
| 30.84@478000: | No $\mathcal{E}$ (GAN) |
| 30.67@492000: | No $\mathcal{E}$ (GAN) |
| 30.50@500000: | Base |
| 27.91@462000: | High Res $\mathcal{E}$ |
| 27.74@498000: | High Res $\mathcal{E}$ |
| 27.50@473250: | High Res $\mathcal{E}$ |

Figure 12: Image generation learning curves for several of the ablations in Section 3 (main text), including a comparison of BigBiGAN to standard GAN. Legend entries correspond to the following rows in Table 1 (main text): *Base*, *No $\mathcal{E}$ (GAN)*, and *High Res $\mathcal{E}$ (256)*.

## Appendix D  Learning curves

In this Appendix we present learning curves showing how the image generation and representation learning metrics that we measured evolve throughout training, as a more detailed view of the results in Section 3 (main text), Table 1 (main text). We include plots for the following results:

- Image generation (Figure 12)
- Latent distribution $P_{\mathbf{z}}$ and stochastic $\mathcal{E}$ (Figure 13)
- Unary loss terms (Figure 14)
- $\mathcal{G}$ capacity (Figure 15)
- High resolution $\mathcal{E}$ with varying resolution $\mathcal{G}$ (Figure 16)
- $\mathcal{E}$ architecture (Figure 17)
- Decoupled $\mathcal{E}/\mathcal{G}$ learning rates (Figure 18)

**Fréchet Inception Distance (FID)**

31.63@498500: Uniform $P_z$
31.40@496500: Base
31.21@479000: Deterministic $\mathcal{E}$
31.13@497250: Uniform $P_z$
31.07@498500: Uniform $P_z$
30.97@482750: Deterministic $\mathcal{E}$
30.85@499000: Base
30.75@492750: Deterministic $\mathcal{E}$
30.50@500000: Base

**Top 1 Classification Accuracy (%), Val. (Cls.)**

48.29@461000: Base
48.06@477250: Base
47.96@497250: Base
47.26@491500: Deterministic $\mathcal{E}$
47.16@468750: Deterministic $\mathcal{E}$
46.48@497500: Deterministic $\mathcal{E}$
46.12@497750: Uniform $P_z$
45.33@499500: Uniform $P_z$
43.88@491000: Uniform $P_z$

**Relative L1 Reconstruction Error (%)**

73.46@476000: Uniform $P_z$
73.31@438750: Uniform $P_z$
72.62@490500: Uniform $P_z$
72.20@484250: Base
71.80@495250: Base
71.42@461000: Deterministic $\mathcal{E}$
71.30@441750: Deterministic $\mathcal{E}$
71.14@408750: Base
71.01@417250: Deterministic $\mathcal{E}$

Figure 13: Image generation and representation learning curves for the latent space variations explored in Section 3 (main text). Legend entries correspond to the following rows in Table 1 (main text): *Base*, *Deterministic* $\mathcal{E}$, and *Uniform* $P_{\mathbf{z}}$.

**Fréchet Inception Distance (FID)**

- 43.64@481750: No Unaries
- 42.70@418500: No Unaries
- 41.49@475000: No Unaries
- 40.32@460250: z Unary Only
- 39.02@475250: z Unary Only
- 38.39@493250: z Unary Only
- 32.07@477250: x Unary Only
- 31.79@493750: x Unary Only
- 31.40@496500: Base
- 31.31@497000: x Unary Only
- 30.85@499000: Base
- 30.50@500000: Base

**Top 1 Classification Accuracy (%), Val. (Cls.)**

- 48.29@461000: Base
- 48.10@457250: z Unary Only
- 48.06@477250: Base
- 47.98@482000: x Unary Only
- 47.96@497250: Base
- 47.82@497750: z Unary Only
- 47.76@492250: x Unary Only
- 47.66@493750: No Unaries
- 47.48@496000: x Unary Only
- 47.42@490250: z Unary Only
- 46.93@498250: No Unaries
- 45.53@494500: No Unaries

**Relative L1 Reconstruction Error (%)**

- 75.97@402500: z Unary Only
- 74.01@493750: No Unaries
- 73.48@458750: z Unary Only
- 73.17@451250: z Unary Only
- 72.73@462250: No Unaries
- 72.20@484250: Base
- 71.93@414000: No Unaries
- 71.80@495250: Base
- 71.33@403750: x Unary Only
- 71.14@408750: Base
- 69.48@348250: x Unary Only
- 69.43@486500: x Unary Only

Figure 14: Image generation and representation learning curves for the unary loss component variations explored in Section 3 (main text). Legend entries correspond to the following rows in Table 1 (main text): *Base*, **x** *Unary Only*, **z** *Unary Only*, and *No Unaries (BiGAN)*.

Figure 15: Image generation and representation learning curves for the $\mathcal{G}$ size variations explored in Section 3 (main text). Legend entries correspond to the following rows in Table 1 (main text): *Base*, *Small $\mathcal{G}$ (32)*, and *Small $\mathcal{G}$ (64)*.

Figure 16: Image generation and representation learning curves for high resolution $\mathcal{E}$ with varying resolution $\mathcal{G}$ explored in Section 3 (main text). Legend entries correspond to the following rows in Table 1 (main text): *High Res $\mathcal{E}$ (256)*, *Low Res $\mathcal{G}$ (64)*, and *High Res $\mathcal{G}$ (256)*.

## Fréchet Inception Distance (FID)

26.69@942500: High Res $\mathcal{E}$
26.68@969750: High Res $\mathcal{E}$, RevNet ×2
26.60@974000: High Res $\mathcal{E}$
26.59@908250: High Res $\mathcal{E}$, RevNet ×4
26.57@988000: High Res $\mathcal{E}$, ResNet-101
26.41@950500: High Res $\mathcal{E}$, ResNet ×2
26.39@970250: High Res $\mathcal{E}$, RevNet
26.37@971750: High Res $\mathcal{E}$, RevNet
26.25@964250: High Res $\mathcal{E}$
26.18@998750: High Res $\mathcal{E}$, RevNet

## Top 1 Classification Accuracy (%), Val. (Cls.)

58.68@869250: High Res $\mathcal{E}$, RevNet ×4
56.12@992250: High Res $\mathcal{E}$, RevNet ×2
54.40@886750: High Res $\mathcal{E}$, ResNet ×2
53.54@995500: High Res $\mathcal{E}$, ResNet-101
53.08@958750: High Res $\mathcal{E}$
53.07@980750: High Res $\mathcal{E}$
52.55@995250: High Res $\mathcal{E}$
51.97@960250: High Res $\mathcal{E}$, RevNet
51.79@922000: High Res $\mathcal{E}$, RevNet
51.59@976750: High Res $\mathcal{E}$, RevNet

## Relative L1 Reconstruction Error (%)

71.37@958750: High Res $\mathcal{E}$
71.26@987250: High Res $\mathcal{E}$
70.99@989250: High Res $\mathcal{E}$, RevNet
70.89@976000: High Res $\mathcal{E}$, RevNet ×2
70.75@989250: High Res $\mathcal{E}$, ResNet ×2
70.72@653750: High Res $\mathcal{E}$, ResNet-101
70.60@800250: High Res $\mathcal{E}$
70.55@997250: High Res $\mathcal{E}$, RevNet ×4
70.26@999750: High Res $\mathcal{E}$, RevNet
70.00@993750: High Res $\mathcal{E}$, RevNet

Figure 17: Image generation and representation learning curves for the $\mathcal{E}$ architecture variations explored in Section 3 (main text). Legend entries correspond to the following rows in Table 1 (main text): *High Res $\mathcal{E}$ (256)*, *ResNet-101*, *ResNet ×2*, *RevNet*, *RevNet ×2*, and *RevNet ×4*.

Figure 18: Image generation and representation learning curves showing the effect of decoupling the $\mathcal{E}$ and $\mathcal{G}$ optimizers to train $\mathcal{E}$ with $10\times$ higher learning rate. Legend entries correspond to the following rows in Table 1 (main text): *High Res $\mathcal{E}$ (256)*, *ResNet ($\uparrow \mathcal{E}$ LR)*, *RevNet $\times 4$*, and *RevNet $\times 4$ ($\uparrow \mathcal{E}$ LR)*.

## Footnotes

[1]Preprocessing code from the TensorFlow ResNet TPU model: https://github.com/tensorflow/tpu/tree/master/models/official/resnet.