[Reviews · NeurIPS 2019]

Reviewer 1



Update: I read the author responses and the other reviews. Overall I saw nothing that substantiated changing my score: reproducability / accessibility is an important component of impact to the community. I acknowledge the importance towards representation learning, which was one reason for my original score. ------ Overall the paper is very well written and technically correct. It was enjoyable to read. However, the contributions are more or less incremental and this work to me represents a technical report / demo on what happens if you scale up the architecture of BiGAN. It is incremental as it takes well-established ideas (BiGAN + BigGAN) and combines them in a straightforward way that would be only available to those with the most compute. Making models much much bigger has become popular lately. BERT showed that using a very large transformer on a masked language model achieves SOTA results on important NLP benchmarks (model / approach novelty and SOTA). BigGAN showed that very large GANs can be used to generate high-resolution datasets reliably (high impact as this was the first instance of full imagenet generation with GANs). GPT-2 showed that large RNN models along with lots of data could generate uncannily realistic paragraphs (high impact as this was the first instance of language generation of this sort at this scale). All of these works were important as they challenged expectations of what was possible as we scale neural networks up. And now there's BigBiGAN. Which is mostly just as the title describes it: Big + BiGAN (or BigGAN + BiGAN if you will). BiGAN is important, I suppose, as it demonstrates a particular paradigm of adversarial learning, but its significance isn't really well established in this work beyond "good linear probe classification performance". I would say that the modifications to the discriminator are useful for training BiGANs and worth exploring more. But it's not entirely clear just making BiGAN big is really an important step forward toward answering some fundamental question in ML other than "what if we make it big?". This question used in this way is not very useful to those outside researchers that can actually train these models. As a tech demo, this paper is excellent. However, I can't imagine it having much significance in the community other than hitting a benchmark as most researchers cannot reproduce these results. So the reproducability score on this paper is naturally low (though the techniques are described very clearly). Why not provide a more accessible demo across some of the models being compared with smaller datasets / models (i.e., reimplement some of those models and train them on similar settings)? Everything above sounds very negative, but overall the paper is very good and I think it deserves to be accepted. It would also help though if the representation learning / self-supervised works mentioned / compared to weren't just from DeepMind / Google, as there are many other relevant works out there: e.g., Noise as Targets (FAIR), Deep Infomax (MSR), and Invariant Information Clustering (Oxford) to name a few. One question: how do the generative results compare to VQ-VAE-2? Do you think VQ-VAE-2 outperforms yours in classification? ------ One last thing, which *did not* affect my score: The claims of SOTA are already dated, but I understand that the authors did not have access to these works at submission. But I would expect the message to be augmented and these works to be at least mentioned for camera ready, if this paper gets accepted, in order to not mislead (which is important): Big CPC: Data-Efficient Image Recognition with Contrastive Predictive Coding CMC: https://arxiv.org/abs/1906.05849 both of which get scores close to yours without a generator. AMDIM: Learning Representations by Maximizing Mutual Information Across Views which gets much better results than yours (68.1% on imagenet)

Reviewer 2



Originality: This work is a novel combination of two well-known models, namely BiGAN and BigGAN. The main novelty lies in demonstrating the effectiveness of additional unary terms for x and z that were not studied in BiGAN. The related work is well-cited. Quality: The submission is technically sound. The claims are well-supported by extensive and well-considered empirical analysis. There are no theoretical contributions. The authors are careful and honest in their evaluation. Clarity: The writing is very clear. Pedagogically, it would be helpful to readers who are not already familiar with BiGAN to read an intuitive explanation of how the forms of L_EG and L_D implement their cross-purposes. Significance: BigBiGAN the natural pushout of the BiGAN <- GAN -> BigGAN triad, so this work is more inevitable than it is surprising. However, the work remains a significant contribution as the investigation is exemplary in its clarity and thoroughness, the methodology is advanced by unary terms, and the performance is SOTA with respect to standard benchmarks in the community. The semantic learning apparent in the images, hypothesized to result from an implicit form of reconstruction error, is especially exciting as an area of future research.

Reviewer 3



Though there isn't much new technical content in this paper, I think it provides impressive empirical motivation for continuing to explore generative models as mechanisms for representation learning. The authors demonstrate the power of ALI/BiGAN as a technique for representation learning by reimplementing and re-tuning the core idea using the biggest, strongest current GAN model (i.e., BigGAN). The result is a model which matches the representation learning performance of concurrent self-supervised methods, while also improving on the prior state-of-the-art for unsupervised GAN-type image generation (by beating BigGAN). The empirical results are impressive. My main concern is that the work is generally unreproducible due to massive computation costs. Nonetheless, I think the work is worth publishing since it will encourage people to keep developing generative approaches to representation learning, some of which will hopefully be more efficient and practical to work with. The paper was clear and easy to read, which is a plus. Post Rebuttal: I found the authors' response to my statements about model size a bit misleading. I think their main result is the "state-of-the-art on ImageNet" result, and the encoder they use for this result is the RevNet x16 model from the CVPR 2019 "Revisiting Self-Supervised Visual Representation Learning" paper. That model is massive, and has perhaps 10x the compute cost of the ResNet50 to which they refer in the rebuttal. Additionally, training the encoder requires training the rest of BigGAN too, which isn't exactly fast and cheap.

[Author Response · NeurIPS 2019]

We thank the reviewers for their thoughtful feedback. We are grateful that all the reviewers recommended the paper for acceptance, and noted clear presentation, strong empirical results, and thorough ablations.

R1 questioned whether BigBiGAN truly represents an "important step forward toward answering some fundamental question in ML" rather than merely an empirical contribution or "tech demo". In our view, the question of whether generative models can learn interesting representations and be applied beyond generation is rather fundamental. We believe our work makes a strong case that unsupervised generative models of images are capable of learning interesting semantics which are practically useful for downstream tasks. With the exception of our work, the unsupervised representation learning field is currently dominated by methods based on self-supervision. Showing that state-of-the-art (at the moment of submission) results can be achieved using an entirely different family of methods is likely to be impactful, and will further motivate the community (especially those working on generative models) to explore representation learning applications further (R3 also mentioned this as a benefit). Beyond improving empirical results, both R1 & R2 noted our improved joint discriminator with unary loss terms, with R2 describing this as a "well-motivated methodological contribution".

On R1's specific concern that we only demonstrate "good linear probe classification performance", we now have additional results which we'll include in a future revision showing solid classification performance (43.3% top-1 accuracy) from non-parametric $k$-nearest neighbors ($k$-NN) classifier in the learned representation space.

R1 & R3 suggested adding demonstrations in smaller settings. We haven't experimented with smaller datasets (e.g. MNIST, CIFAR) mainly because the prior work we primarily build upon (ALI, BiGAN) has addressed these settings, and we expect little to be gained there. Note also that among recent work in unsupervised representation learning for images, ImageNet-focused evaluations are quite standard and small low-resolution datasets are seldom considered; there is simply not enough semantic "juice" in these tiny datasets to enable interesting representation learning. On model size: for the encoder, most of our experiments use the standard ResNet50 architecture – on the smaller end of widely-used models in visual recognition today – and for the generator, our ablations show that we can't afford to reduce its size much without significant cost to representation learning performance. That said, we do acknowledge that training these models involves significant computational costs and consider improved scalability an important goal for future research.

R2 & R3 suggested providing code for reproducibility. In a future revision, we are planning to release pre-trained models, and ensure that all training hyperparameters are fully specified.

R2 suggested visualizing iterated reconstruction results, where the reconstructed input image is passed through reconstruction again multiple times. This is an interesting experiment, and in fact we were curious about this as well and have created visualizations which we will add to a future revision. We did not observe an exact fixed point, but did notice that the process goes through stages, each of which produces images semantically similar to each other, while the images are quite different across the stages. We hypothesize this is due to the iterated reconstruction eventually reaching a point outside of the natural image distribution the model has been trained on, which leads to semantically different reconstructions at the next step. R2 also suggested an intuitive explanation of the losses – we will add this in a future revision as well.

R1 suggested adding comparisons to other methods, including results published after the submission deadline. We will include the results from concurrent work in a future revision of the paper. Regarding the results included in the submission, we attempted to include in Table 2 all recent competitive results on ImageNet from unsupervised approaches we were aware of, as well as a result from the original BiGAN due to its relevance to our method.

Finally, R1 asked for comparisons with VQ-VAE-2 on generation and representation learning. For ImageNet generation, BigBiGAN is trained unconditionally (without class information), while VQ-VAE-2 is class-conditional, so it's not fair to compare the two. Generally, BigBiGAN generators underperform state-of-the-art class-conditional generative models (BigGAN, VQ-VAE-2, etc.), which is expected, since our generator does not receive information about which class it should generate. VQ-VAE-2 did not report representation learning results (unsupervised or otherwise).

[Meta-Review · NeurIPS 2019]

A algorithmic scaling-up paper for representation learning with a better generator by combining BiGAN and BigGAN. Well-explored ablation studies showcasing the efficacy of the approach. While the reviewers have raised concerns of compute time, I think this paper needs sharing broadly. I would suggest authors incorporating suggestions from the reviews to strengthen the paper; this includes visualizations of iterative generation and additional results.